# Promoting energy conservation through attention control and construct activation: A field test at a campus laundry

Steven G. Buzinski[ID][1]*, Tatum A. Jolink[2°], Olivia C. Smith[1‡], Amit M. Ariely[1‡], Paschal Sheeran[1°]

**1** Department of Psychology and Neuroscience, University of North Carolina at Chapel Hill, Chapel Hill, North Carolina, United States of America, **2** Department of Psychology, University of Michigan, Ann Arbor, Michigan, United States of America

° These authors contributed equally to this work.
‡ OCS and AMA also contributed equally to this work.
* buzinski@email.unc.edu

## Abstract

Increasing the frequency of laundering in cold water has been demonstrated to reduce energy consumption, but heated water laundering rates remain high. This is particularly true in wealthy, western nations. This paper describes an individual-level, situated intervention utilizing attentional control and construct activation to motivate cold water laundering. In Study 1, we report the first direct observational study of laundering in a field setting, finding that (a) only 21% of laundry loads were washed with cold water, and (b) behavioral, normative, and control beliefs were associated with laundering in cold water. In Study 2, we report an experimental field test of an intervention designed to increase rates of cold-water laundering. The intervention significantly increased cold water laundering (57%) compared to a control condition (25%). We also tested mediators of the intervention and laundering behavior association and observed that control beliefs fully mediated the relationship. We discuss the theoretical and practical contribution of these studies to the environmental psychology and motivational science literatures.

## Introduction

A report by the American Psychological Association's Task Force on the Interface Between Psychology and Global Climate Change (2011) stated that psychological science could contribute more to the fight against climate change by designing more effective individual-level interventions. This project describes one such intervention. Through the strategic direction of attention and activation of psychological constructs associated with environmental action, our intervention increased the frequency of an important energy-saving behavior, laundering in cold rather than heated water.

**Data availability statement:** Yes - all data are fully available without restriction; All relevant data can be found at the following: https://osf.io/bfywk/.

**Funding:** The author(s) received no specific funding for this work.

**Competing interests:** The authors have declared that no competing interests exist.

### Laundering behavior and energy consumption

Evidence from social and environmental psychology has demonstrated across multiple domains that individual-level interventions can be used to increase energy-conserving behaviors [1–3]. For example, Varotto and Spagnolli [3] meta-analyzed 36 studies reporting 70 interventions with the goal of increasing recycling behavior. They found that interventions based on social modelling ($d = .40$) and, particularly, environmental alterations ($d = .73$) were effective at increasing participants' recycling behavior. Compared to recycling, however, there is relatively little cultural awareness of, and research into, the way people launder clothing, and how different forms of laundering affects their energy usage. It has, however, recently garnered more attention and focus of study.

In many countries, people now own more articles of clothing than ever before and correspondingly launder more often [4]. In the United States of America alone, 59.8 million households wash one to four loads of laundry per week, and 26.9 million households wash five to nine loads per week, according to the U.S. Energy Information Administration [5]. In the United Kingdom, the amount of energy used to wash and dry clothing increased by more than 100% from 1970 to 2012 [6]. These trends are consequential to the environment because life cycle studies (i.e., assessments of the environmental impact associated with the stages of a product's life, from raw material to destruction) show that energy consumption is highest during the "consumer use" period for both clothing and washing machines [7], making laundering a substantial environmental impact [8]. To wit, washing machines use 341 kWh of electricity per household per year, producing up to 0.24 metric tons of greenhouse gas emissions [9]. The extant research makes it clear that the increasing rate of laundering is problematic, but one viable strategy to mitigate this source of energy consumption is to reduce the temperature at which people launder.

When a recent analysis of laundering norms compared the environmental impacts of laundering in China, Germany, Japan, the UK, and the USA, it found that Japan used the least amount of energy due to primarily laundering in unheated water [10]. The reason that using unheated water leads to such an energy savings is because a significant proportion of the energy consumed while laundering is used to heat the water for warm or hot temperature cycles. Easter et al. [11] found that in a standard load of laundry approximately 50% of the energy consumed could be attributed to heating the water. Similarly, Laitala et al. [4] report that washing clothes at 30°C uses 29% less energy than washing them at 40°C, and 58% less energy than a 60°C wash.

An article published by National Geographic in 2019 reported that the energy cost to heat the water during a wash cycle was greater even than the energy required to produce the detergent used and all packaging material [12]. The same article reported that in Europe up to 60% of the greenhouse gas emissions from laundering comes from water heating. These findings reaffirm that encouraging people to launder in cold water is a strategy that has the potential to have a substantial environmental impact. Shifting laundering norms toward using more cold-water cycles would be particularly helpful in high-use countries like the USA [10]. According to a report

published by The Sustainability Consortium, if all households in the United States washed one load of laundry per week in cold rather than warm or hot water for one year, electricity use could be reduced by 2,007 million kWh, natural gas use by 166 million therms, and greenhouse gas emission by 2.3 million metric tons. These energy savings are approximately equal to Americans driving 5,498 million less miles [9].

Despite the weight of evidence for the environmental value of laundering in cold water, several factors contribute to the continued use of warm and hot water cycles. One such factor is the belief that higher water temperatures clean more effectively. Carrico, Spoden, Wallston, and Vandenbergh [13] found that 69% of participants incorrectly perceived that handwashing in hotter water temperatures kills more germs compared to cooler water temperatures. Similarly, Laitala et al. [4] found that 92% of participants trusted the hot water cycle to effectively clean a soiled shirt, whereas only 51% trusted the cold water cycle to do the same. These misperceptions persist even though modern detergents clean just as effectively in cold water. Moreover, people are often unaware of amount of energy used in heated water laundering [4,9], which means that pro-environmental goals would not become more cognitively accessible in the laundering situation itself, and would instead require some form of situational cue to become activated.

For interventions to effectively overcome the default of hot water washing and motivate people to use cold water instead, strategies should be grounded in theories of motivated behavior. Such strategies would ideally direct attention to, and activate favorable psychological constructs associated with, pro-environmental behaviors (i.e., laundering in cold water).

## Attention, construct activation, and motivated behavior

The Attention-Readiness-Motivation framework is a recent formulation of motivated behavior that can be used to inform behavior change interventions. Whereas other motivational theories focus predominantly on the desirability and feasibility considerations that determine the value of an action [14,15; see 16 for a review], the ARM framework proposes that endogenous attention is a necessary precursor of the effect of these valuations. Endogenous attention refers to the orienting of an individual's attention towards outcomes associated with an environmental stimulus. Attention to, and the valuation of the outcome of engaging a stimulus, then, operate in concert to generate the motivation to enact a corresponding behavior. Endogenous attention serves as a psychological barrier to entry that, if focused, unlocks the influence of an action's value to the individual, making it, if sufficiently valued, an accessible means of goal pursuit. Of the possible behaviors that could serve to accomplish one's goals in any given situation, it is the behavior that attracts attention and is sufficiently positively valued that is likely to be initiated.

Suri and colleagues [17–18] conducted several studies exploring the relationship between endogenous attention and motivated behavior. In one such study the researchers manipulated the presence of a free-standing sign in a train station where commuters had to choose whether to take the stairs (a means to the goal of being healthy) or an escalator (a habitual behavior). The conditions included signs that read "Stairs" or "Stairs or Escalator?" and a no-sign control. When attention was directed to the opportunity for goal-congruent action, through either sign condition, a greater proportion of the commuters chose to take the stairs instead of the escalator, compared to both a no-sign control condition and a sign that did not draw participants' attention to the stairs vs. escalator decision ("Have a good day!") [18]. Suri et al.'s [18] research thus provided evidence that directed attention can increase motivation by making the opportunity for goal pursuit more salient, unlocking the valuation of that goal-directed behavior, and increasing the likelihood of its performance.

Psychological construct activation (e.g., activating attitudes), as opposed to construct change (e.g., increasing attitude strength), has also been demonstrated to be an effective mechanism of behavior change. Construct change interventions are those in which interventions affect behavior through the direct modification of outcome-relevant thoughts, values, and goals, whereas construct activation interventions raise the cognitive accessibility of outcome-relevant thoughts, values, and goals in critical situations [19]. Sheeran et al.'s [1] research concluded that strategies such as nudges, cues, and goal priming are interventions that operate by doing just that – increasing construct accessibility, rather than changing the value

of those constructs. This form of intervention is particularly efficacious when thoughts, feelings, and goals are already consistent with the desired behavior but are not otherwise activated during goal-relevant moments.

The effectiveness of construct activation interventions is demonstrated by Verplanken & Holland [20], who found that priming environmental values led to environmentally-friendly consumer decision-making. In a dual study paradigm, participants were initially presented with a person perception task in which 20 values of a target person were described. In the experimental condition, 12 of their values were environmental (e.g., preserving nature, living in a healthy place) whereas in the control condition, zero of their values were environmental. Then, in what was described as an unrelated task, participants evaluated a series of television sets that varied on seven dimensions, one of which was their environmental impact. Primed participants were significantly more likely to decide to purchase a television set that rated high in environmental friendliness than were non-primed participants. Importantly though, this outcome occurred only when environmentalism was a value that the participant had strongly endorsed during a pre-testing period. As discussed above, when thoughts, feelings, or in this case, values, are already consistent with a desired outcome, raising their cognitive accessibility in a critical situation will motivate outcome-consistent behavior. Our intervention used situational construct activation, as well as attentional control, to motivate an environmentally-friendly behavior, laundering in cold water.

### The present research

The foregoing discussion suggests that utilizing attention control and construct activation will provide a strong conceptual basis for an intervention designed to motivate the environmental behavior of laundering in cold water. Thus, we sought to accomplish two objectives in the current project: (1) examine laundering behavior in a naturalistic field setting, and (2) reduce energy consumption on a college campus through the application of a situated intervention designed to increase the use of cold-water laundering options. In Study 1, we used naturalistic observation and self-report measures to assess the frequency of, and psychological constructs associated with, laundering in cold water. This study was essential for two reasons: (a) to establish a baseline of cold-water laundering rates before implementing an intervention, and (b) to identify the psychological constructs associated with participants' laundering behavior. Whereas prior research has demonstrated that directing attention to decision points and activating supporting cognitive constructs can promote goal-congruent actions, laundering in cold water may have been a novel behavior for our participants. Therefore, we needed to assess the key determinants of their decision-making process in order to design a targeted intervention. In Study 2, we conducted a field experiment in a campus laundry facility, drawing upon the findings from Study 1 and applying the principles of attention control and psychological construct activation to facilitate behavior change and reduce energy consumption.

## Materials and methods

### Study 1: Rates and predictors of laundering in cold water

College campuses, as major consumers of both water and energy for laundering, present a rich opportunity to examine the frequency at which people use cold water to wash their laundry, and why people choose (not) to wash with cold water. To avoid biases associated with retrospective self-reports, we undertook the first field study using behavioral observation, as well as in-the-moment survey techniques, to measure cold water laundering frequency and the psychological constructs related to laundering in cold water. We hypothesized that three psychological constructs would predict laundering with cold water [cf. 14]: normative beliefs (expectancies about rates and approval of its use on campus), behavioral beliefs (expectancies about the energy consequences of laundering in cold water), and control beliefs (expectancies concerning one's capacity to undertake cold water laundering).

**Method. Participants:** A total of 218 undergraduate students at a large, research-intensive university in the Southeastern United States participated in the study between the dates of 01 February 2019 and 08 February 2019. The study was approved by the hosting university's Institutional Review Board, project #19–0251. No minors were allowed

to participate. Students were recruited as they washed laundry at the largest student residence hall on campus. Once they agreed to participate in the study, students were consented through an electronic consent form on Qualtrics. Of the participants who provided consent, 61.9% identified as female, 32.1% identified as male, and 1.4% (3 participants) self-identified as "other" or non-binary. In all, 92.2% of the sample were first-year students, with a median age of 18 ($M = 18.5$, $SD = 0.84$). Additionally, 57.3% of the participants identified as White, 8.3% as Black, 3.2% as Hispanic or Latinx, 8.7% as East Asian, 5.0% as South Asian, 11.5% as Biracial, and 1% as another race. Demographic information was missing from eleven participants. Twelve other participants had to be excluded from the analyses due to experimenter documentation error.

**Procedure:** We collected data over eight consecutive days. Experimenters surreptitiously observed college students washing laundry in the laundry room of the largest dormitory on campus from the hours of 9:00 AM until 10:00 PM. The room contained ten washers, arranged in two rows of five, running from the right wall to the left side of the room. This divided the room into two sections, with five washers on each side, numbered sequentially from one to ten. A lounge room, with a large pane-glass window looking into the main laundry area, was adjacent to the laundry room and served as an observation room from which the experimenters could unobtrusively watch as students entered and began their laundry.

For each prospective participant, experimenters documented the date, washing machine used (e.g., 1–10), and, most importantly, wash cycle selected. If the prospective participant washed more than one load of laundry, and therefore used two or more washers, each load was recorded as a separate observation. Once prospective participants had selected a wash cycle, the experimenter approached and introduced the second part of the study. Participants were told that the study was being conducted on students' "attitudes about laundry," and asked if they would be willing to complete a short, 5-minute online survey to assess their laundry thoughts and habits. More than 93% of the observed students consented to participate in the survey. After doing so, they completed the electronic survey on either their personal device or a laptop provided by the experimenter. Upon finishing the survey, participants were thanked for their time.

**Materials. Laundry questionnaire:** Participants first responded to two questions about their laundry. The first was, "what kind of laundry load are you currently washing?" Answer options included, a) Whites, b) Colors, c) Delicates, d) Towels, and e) Mixed. The second was the self-reported question, "which wash cycle did you select?" Answer options reflected the preset options on the washing machine.

**Psychological constructs associated with laundering in cold water:** Participants then completed measures of behavioral ($r = .34$), normative ($\alpha = .62$), and control beliefs ($\alpha = .83$) [21]. The scale can be found in Table 1.

**Table 1. Items and reliabilities of behavioral, normative, and control beliefs in Study 1.**

| Factor | Items |
|---|---|
| Behavioral Beliefs | Cold water washing uses less energy than warm water washing. |
| | Warm water washing is more economical than cold water washing.* |
| Normative Beliefs | Most [University] students use cold water when washing their clothes. |
| | Most [University] students think it is important to use cold water washing. |
| | Most [University] students use warm water washing when washing their clothes.* |
| | Most [University] students believe that cold water washing doesn't clean clothes as well as warm water washing.* |
| Control Beliefs | Using cold water washing when doing my laundry is… <br> Easy - - - Difficult <br> Confident - - - Unconfident <br> Under my control - - - Not under my control |
| | I know which button/setting on the washing machine uses cold water washing. |
| | I feel capable of using cold water washing. |
| | I am discouraged from using cold water washing because I feel I do not know how to.* |

*The item was reverse coded.

**Past behavior:** Participants were also asked about their past laundry behavior with the prompt, "How frequently you have used cold water to wash during this academic year?" They answered on two seven-point scales, one ranging from *never* to *often* and the other from *never* to *every laundry load* (*r*= .92, *p*< .001).

   **Analysis plan.** We used a binary logistic regression to predict laundering in cold water from each factor (i.e., behavioral beliefs, normative beliefs, and control beliefs). The binary outcome was using a cold-water cycle (i.e., selecting the "brights" button on the washing machine settings panel) or not (i.e., selecting any of the other five wash settings). Additionally, we conducted linear regressions to predict past cold-water laundering behavior using the same predictors.

### Study 2: Test of an attention control and construct activation

   **Situated intervention to increase cold water laundering.** We conducted a second, experimental field study in which we used the information gleaned from Study 1, along with the theoretical principles of attentional control [22], and psychological construct activation [19], to develop an energy-saving behavioral intervention. First, as laundering is habitual (i.e., repetition of the same sequence of actions similar contexts), we felt that attention had to be drawn to the choice point (i.e., selecting a wash cycle) to disrupt habitual responding. Thus, the intervention was deliberately designed to attract attention with a salient, environmental cue (Fig 1). The cue was an arrow, 8 inches by 4.5 inches in size, containing text and the colors associated with the university's sports teams, and was placed in a conspicuous position on the front of the laundry machine. Second, following the research of Sheeran et al. [19], we raised the accessibility of outcome-supporting psychological constructs in the critical situation (i.e., the moment of selecting a wash cycle). We did so in the following ways. First, we placed three messages in bold text on the body of the arrow. The message at the top of the arrow raised the accessibility of normative beliefs by describing the degree of support for environmentalism on campus (i.e., "Most [name of university mascot] want to fight climate change"), the message in the middle of the arrow raised the accessibility of behavioral beliefs with a statement about the energy consumption of heated water cycles ("90% of the energy used by a washing machine is to heat the water"), and the message at the bottom of the arrow raised the

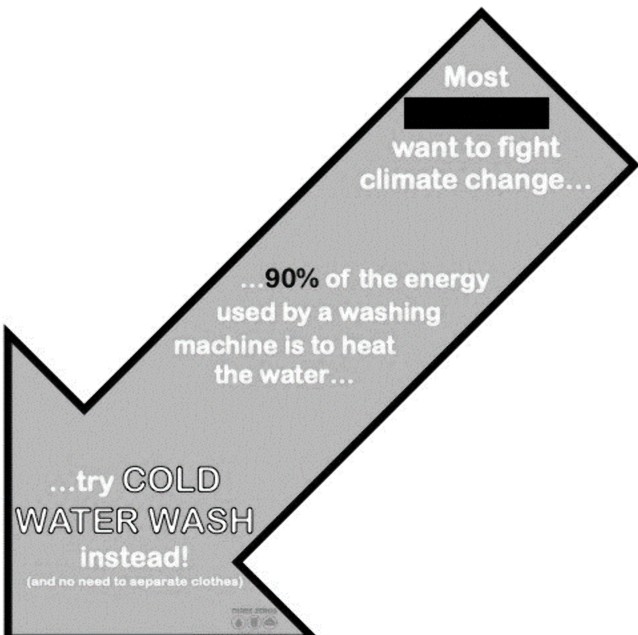

**Fig 1. Image of the intervention condition, which was placed on each washing machine.**

accessibility of control beliefs by suggesting an easy behavioral response ("[T]ry cold water wash instead! [and no need to separate clothes]"). Additionally, we placed the physical tip of the arrow such that it pointed to the cold-water cycle button, which was also intended to increase perceived behavioral control.

We hypothesized that the intervention condition would produce higher rates of laundering in cold water compared to a no-intervention control condition. We also sought to test the role of our conceptual variables, attention and construct activation, in the study. In particular, we tested whether (self-reported) attention, behavioral beliefs, normative beliefs, or control beliefs would mediate the relationship between experimental condition and laundering behavior.

**Method. Power analysis:** The power analysis assumed that the treatment would cause more than one-half of participants (i.e., 51%) to use a cold-water cycle. We used the rate of cold-water laundering observed in Study 1 (21%) as the estimate for the control condition. Analyses using G*Power indicated that 78 participants ($n = 39$ per condition) should enter the trial to have 80% power to observe a significant difference at alpha $< .05$ (two-tailed).

**Participants:** A total of 83 undergraduates at a large, research-intensive University in the southeast United States participated in Study 2 between the dates of 04 March 2019 through 22 March 2019. The study was approved by the hosting university's Institutional Review Board, project #19–0251. No minors were allowed to participate. Students who had participated in Study 1 were excluded from Study 2. As in Study 1, students in Study 2 were observed in the laundry room of the largest campus residence hall. Once they agreed to participate, students were consented using an electronic consent form on Qualtrics. The majority of the sample identified as female (67.9%), were in their first year of college (94.0%), and had a median age of 19.0 ($M = 18.55$, $SD = 0.60$). Participants identified as white (63.1%), Black or African American (6.0%), Hispanic or Latino/a/x (4.8%), East Asian (10.7%), South Asian (3.6%), American Indian or Alaskan Native (2.4%), biracial (7.1%), and other (2.4%).

**Procedure:** The study took place over two non-consecutive weeks in March 2019, from the hours of 9:00 AM to 11:00 PM, in the same laundry room as used in Study 1. Participants were randomly assigned to the control or intervention condition. Following Suri and Gross [17], assignment was based on temporal intervals. At the beginning of each 2-hour block, the pre-designated condition was placed on the washing machines by the experimenter. We used this design because the physical arrangement of the laundry room – with two rows of five washers situated back-to-back – made it impossible to randomly assign individual participants. Specifically, with the washers right next to each other, it was not possible to place different conditions on each one, for participants would have easily seen the distinct instructions (or lack thereof) between the machines. Doing so would have pierced the random assignment.

Experimenters surreptitiously observed and recorded the wash cycle each participant selected, as well as the time, date, specific machine used, and experimental condition. After the participants selected a wash cycle, we approached and asked them to complete an online survey regarding their "attitudes about laundry." Importantly, the experimenter positioned the computer on which the survey was displayed to ensure that participants had their backs to the washing machine and could not see the arrow intervention while completing the survey.

**Materials. Experimental manipulation:** We designed Study 2 to assess the impact of the attention control and construct activation-based intervention against a control condition on college students' cold-water laundering. In the control condition, the laundry room's existing signage and washing machines were left as is, nothing was added nor removed. The intervention was an 8.0 x 4.5-inch light blue arrow that was placed directly on all of the washing machines in the laundry room. The arrow was placed so that it curved over the top and down the front of the machines, culminating in the tip of the arrow pointing to the cold-water option on the washing machine's panel. The face of the arrow contained the normative (i.e., "Most [NAME OF UNIVERSITY MASCOT]s want to fight climate change"), behavioral (i.e., "90% of the energy used by a washing machine is to heat the water"), and control belief (i.e., "[T]ry cold-water wash instead! [and no need to separate clothes]") messaging.

We used "treatment as usual" or "usual care" as the control condition (see Freedland et al. (2019). That is, the laundry room signage required by the Housing Authority remained in place and no additional signage concerning cold-water

laundering was provided. This meant that the laundry room was the same as it would have been if no intervention was tested, and was equivalent to the physical setting tested in Study 1.

**Measures:** In addition to the primary dependent measure (i.e., laundering with a cold-water cycle), we included a self-report measure of attention to assess the extent to which participants attended to the intervention during the experimental period. We did this in order to test its mediating role in the intervention-behavior relationship. Specifically, participants responded to the question, "Did you see a sticker or anything added to the washing machine?" Answer options ranged from 1 (definitely no) to 5 (definitely yes). Additionally, we measured behavioral ($r = .41$), normative ($\alpha = .68$), and control beliefs ($\alpha = .84$)using the same items as Study 1.

**Analysis plan.** We used binary logistic regressions to test our primary hypothesis, that the intervention would cause more cold water laundering than the control condition. We also conducted independent samples t-tests to examine the influence of the intervention on attention. Finally, we conducted mediation analyses to test whether attention or construct activation mediated the association between intervention condition and laundering in cold water.

## Results and discussion

### Laundering in cold water frequency in Study 1

We observed 376 instances of laundering over the course of data collection for Study 1. Of those laundry loads, 78 (20.74%) were washed with cold water whereas 298 (79.26%) used a heated water cycle.

**Constructs related to observed and self-reported cold-water laundering.** All measured constructs were significantly associated with laundering in cold water, both observed and self-reported (Table 2). Control beliefs – or the extent to which participants felt that using a cold-water laundering cycle was easy and under their control – were the strongest predictor of participants' use of cold water for laundering ($OR = 4.27$, $p < .001$) and explained the greatest amount of variance in their behavior (observed behavior, Effron's R-squared = .28; past behavior, R-squared = .31). Table 2 displays the results of the adjusted binary logistic regression and linear regression models for each construct.

This is the first study of which we are aware that directly observed laundering behavior on a college campus. We observed students washing 376 loads of laundry and found that only 21% of those loads were laundered in cold water. We also found that behavioral beliefs, normative beliefs, and control beliefs were associated with cold water laundering, with odds ratios indicating 86%, 66%, and 327% increases, respectively, for each one standard deviation increase in the predictors.

Study 1 provides critical information about the prevalence of laundering in cold water on a college campus, as well as the psychological constructs related to its use. The findings indicate that there is considerable scope for behavior change (79% of loads were washed using a heated water cycle), and demonstrate that behavioral, normative, and control beliefs are associated with washing laundry in cold water. Accordingly, we decided that it was appropriate to undertake an intervention to increase cold water laundering, and to use stimuli that would activate behavioral, normative, and control beliefs as part of the treatment.

**Table 2. Constructs associated with laundering in cold water (Study 1).**

| Predictor | Observed behavior | | | | | | | Past behavior | | | |
|---|---|---|---|---|---|---|---|---|---|---|---|
| | Mean | SD | r | B | OR | CI | $R^2$ | r | b | SE | $R^2$ |
| Behavioral beliefs | 5.25 | 1.24 | 0.17 | 0.62* | 1.86 | [1.47,2.39] | 0.10 | 0.32 | 0.57* | 0.09 | 0.10 |
| Normative beliefs | 3.59 | 0.96 | 0.14 | 0.51* | 1.66 | [1.24,2.54] | 0.04 | 0.41 | 0.96* | 0.12 | 0.16 |
| Control beliefs | 5.33 | 1.36 | 0.38 | 1.45* | 4.27 | [2.93,6.56] | 0.55 | 0.55 | 0.92* | 0.08 | 0.31 |

*$p < .001$.

## Laundering in cold water in Study 2

The intervention caused a significantly greater amount cold water laundering (57.14%) than a no-intervention control condition (25.00%) (Fig 2). Binary logistic regression provided converging evidence, showing that the probability of using a cold-water laundry cycle increased for participants in the intervention condition compared to participants in the control condition (B = 1.39, SE = 0.48, p < .001). The odds ratio was 4.00 (95% CI = 1.60 to 10.48), meaning that participants in the intervention condition were four times more likely to wash with cold water.

**Mediators of intervention effects on behavior.** Next, we examined potential mediators of the intervention on cold water laundering behavior. We computed the correlations among condition (1 = intervention, 0 = control condition), observed cold water laundering (1 = yes, 0 = no), and the potential mediators (attention and behavioral, normative, and control beliefs). Table 3 indicates that two factors – attention and control beliefs – were associated with both condition and behavior and could thus qualify as mediators. Accordingly, we undertook formal tests of mediation for attention and control beliefs. We used bootstrapping (seed set at 1000 iterations) to calculate the indirect effect along with other paths of the model; separate tests were undertaken for the two mediators. The indirect effect of the intervention on cold water laundering via attention was not significant, B = .07, SE = .089, p = .78, $CI_{95\%}$ = [−0.10, 0.26], indicating that attention did not mediate the intervention effect on behavior.

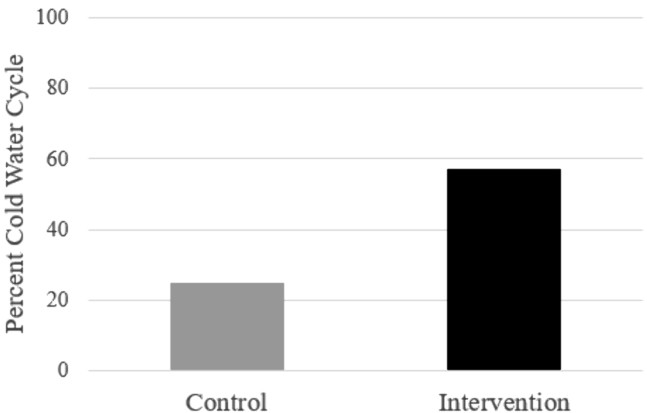

**Fig 2. Effect of the intervention on rate of laundering in cold water.** *p < .05; **p < .01; ***p < .001.

**Table 3. Correlations among condition, cold water laundering behavior, and potential mediators.**

|  | 1 | 2 | 3 | 4 | 5 | 6 |
|---|---|---|---|---|---|---|
| 1. Condition | — |  |  |  |  |  |
| 2. Cold water laundering | 0.33** | — |  |  |  |  |
| 3. Attention | 0.66** | 0.27* | — |  |  |  |
| 4. Behavioral beliefs | 0.09 | 0.16 | 0.09 | — |  |  |
| 5. Normative beliefs | 0.11 | 0.25* | 0.02 | 0.09 | — |  |
| 6. Control beliefs | 0.31** | 0.39*** | 0.38*** | 0.28** | 0.33** | — |

Condition was dummy coded (1 = intervention, 0 = control condition).

*p < .05.

**p < .01.

***p < .001.

Bootstrapped confidence intervals (1000 iterations) supported mediation by control beliefs ([Fig 3]). The intervention pre-dicted stronger control beliefs, $B = .77$, $SEB = .24$, $p = .001$. Controlling for experimental condition, control beliefs were also significantly associated with laundering in cold water, $B = .13$, $SEB = .04$, $p = .001$. Finally, the indirect effect of condition on behavior through control beliefs was significant, $B = .10$, $SEB = .04$, $p = .027$, $CI_{95\%} = [0.04, 0.22]$.

## Discussion

In Study 2 we tested an attention-control and construct activation intervention against an existing-signage control condition on washing clothing in cold (vs. heated) water in a university residence hall laundry. Our findings in the control condition (25.00%) replicated the cold-water laundering frequency observed in Study 1. The intervention, however, significantly increased the rate at which students laundered in cold water (56.76%). Odds ratios indicated that participants in the intervention condition were four times more likely to wash with cold water than participants in the existing-signage control condition.

We also tested multiple potential mediators of the intervention and cold-water laundering relationship. Whereas behav-ioral beliefs, normative beliefs, and self-reported attention to the relevant signage did not mediate the intervention's influ-ence, control beliefs did so. The intervention increased participants' beliefs in their capacity to successfully launder in cold water, which in turn increased the performance of cold-water laundering behavior.

## General discussion

The present research offers one of the first behavioral trials to use a low-cost, situated intervention to improve the energy efficiency of a campus laundry. This project is practically important because increasing laundering frequency, combined with an existing preference for heated water cycles, is causing significant energy waste [9–11,13] and therefore provides an opportunity for energy savings. We tested our ideas across two field studies at a campus laundry at a large university in the southeastern United States of America. First, we naturalistically observed students' laundering behavior to collect data on base rates of cold water laundering and the psychological constructs associated with it (Study 1). We found that approximately one-fifth (21%) of laundry loads were washed using a cold-water cycle, and that behavioral, control, and normative beliefs were associated with laundering in cold water. These findings alone add to the existing body of scientific knowledge on laundering behavior because all previous research on cold water laundering rates used self-report mea-sures rather than direct observation.

Based on the findings of Study 1, and the attention and construct activation intervention literatures [20,22] we designed and tested a situated intervention to increase the proportion of students who laundered in cold water (Study 2). The inter-vention was a physical sign in the shape of an arrow secured directly to the washing machines, with its tip pointed to the cold-water cycle option. As such, our intervention design was consistent with the research of Gross and colleagues [8,22], in that it directed endogenous attention to a relevant choice point in a critical situation. The body of the arrow contained three messages, one each intended to raise the cognitive accessibility of behavioral, normative, and control beliefs. In all, the placement of the arrow drew attention, reading the messages increased the accessibility of relevant psychological constructs, and the tip of the arrow provided a simple behavioral response that was consistent with those constructs. The

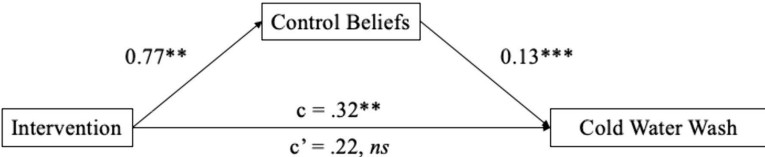

**Fig 3. Control beliefs fully mediate the association between the intervention and laundering in cold water.** *p < .05, **p < .01, ***p < .001.

intervention proved highly effective, significantly increasing the rate of cold-water laundering in the treatment (57%) compared to the control group (25%).

We also explored potential mediators. The intervention did not significantly strengthen behavioral or normative beliefs. Although self-reported attention was significantly greater in the intervention condition and attention was correlated with cold-water laundering, we did not find evidence that attention mediated the condition-laundering behavior association. Rather, we observed that stronger control beliefs explained the association between the intervention condition and laundering behavior. Participants in the intervention condition reported greater efficacy laundering in cold water and thus were more likely to actually launder in cold water. There are two potential reasons why control beliefs formed a significant mediator but attention did not. First, because this was a field experiment attention was measured via self-report. An objective measure of attention, such as observing participants' eye gaze with eye-tracking, would have offered a more precise index of attention allocation and could have altered the findings. Second, the consideration that control beliefs but not attention explained behavior may be consistent with the ARM framework's proposal that endogenous attention is a necessary precursor of the effect of valuations but valuations (in the present case, control beliefs) are the proximal predictor. The signage used in Study 2 involved a simultaneous manipulation of both attention and valuations. Future studies should use a factorial design to afford a stronger test of the individual and combined effects of attention and valuations.

The present findings nonetheless highlight the effectiveness of situated, individual-level interventions in increasing environmentally sustainable behaviors. By structuring the decision environment to encourage energy-efficient choices, such interventions offer a scalable approach to reducing energy consumption in institutional settings like university campuses. To wit, leadership at the campus on which this project was conducted rapidly integrated the arrow intervention into dozens of additional laundry facilities.

Importantly, the focus on increasing cold water laundering to reduce energy waste is not limited to the campus environment. Similar initiatives are being put forward by some of the world's largest consumer product companies, reinforcing the importance of the current work. A recent program, led by Procter & Gamble (P&G), successfully convinced half of National Football League (NFL) teams to agree to wash their uniforms in cold water [23]. The willingness of these teams to launder in cold water not only reinforces the importance of laundering as a source of potential energy savings but also supports the research demonstrating equal cleaning efficacy of cold and heated water cycles. Moreover, industry leaders like P&G, Unilever, and BASF are now primarily investing in detergent formulations that enhance cleaning performance at lower temperatures, a clear indication of a continued national interest in cold water laundering.

The present work does have limitations that should be acknowledged here and addressed in future research. The first limitation is our use of a self-report measure of attention in Study 2. Future research would ideally utilize an objective index of endogenous attention. One possibility would be to replicate Study 2 in a laboratory setting, using a computer simulated version of the intervention conditions while participants wear eye-tracking devices. Another possibility is to conduct another field experiment while taking video recordings of participants' sequence of actions. The videos could then be coded to determine the extent to which participants attended to the intervention. This replication would maintain the ecological validity of Study 2 while also providing an objective measure of attention. Further, Studies 1 and 2 used self-report measures of beliefs, and the behavioral and normative belief scales had moderate scale reliability.

The second limitation to consider is that these studies were conducted in one residence hall laundry room, because we were not able to access other locations. This has the potential to cause two issues. First, it may limit the generalizability of the results due to the demographics of the sample. A majority of the participants (94%) in the intervention study were first year university students. It is possible that this sample was relatively inexperienced in laundering, and thus our findings may be limited to novice groups. Second, because both studies were conducted in the same laundry room, there is the potential that those in the Study 2 sample were made aware of Study 1 by Study 1's participants.

Moreover, although Study 2 had sufficient statistical power to test the impact of the intervention condition, replications of our findings with larger samples in other contexts would be valuable. It was also the case that we were unable to

decompose the influence of the two key components of our intervention – endogenous attention (i.e., the physical arrow and its placement) and psychological construct activation (i.e., the behavioral, normative, and control belief messages on the arrow). Factorial designs could be used in future research in order to formally test the relative, and interactive, influence of each.

Notwithstanding these limitations, the present work makes a valuable contribution to the literature on both energy conservation and motivated behavior. Regarding energy conservation, we report the first behavioral observations of cold-water laundering frequency on a college campus, as well as evidence concerning the psychological constructs associated with it. We also developed and tested an inexpensive, scalable intervention that proved highly effective in increasing an energy- and cost-saving behavior. Regarding motivated behavior, we replicated the endogenous attention findings from Suri and Gross's [22] ARM framework by showing that directing attention to a critical choice point will lead to an increase in belief-consistent behavior. We then extended those findings by identifying that control beliefs mediated the relationship between the attention-based intervention and a target behavior, in this case laundering in cold water. Taken together, these results bridge research from social and environmental psychology to demonstrate that the basic cognitive properties of attention and accessibility can be harnessed to increase energy-conserving behaviors on a college campus.

## Acknowledgments

The authors have no acknowledgments to declare.

## Author contributions

**Conceptualization:** Steven G. Buzinski, Paschal Sheeran.

**Data curation:** Steven G. Buzinski, Tatum A. Jolink, Olivia C. Smith, Amit M. Ariely.

**Formal analysis:** Steven G. Buzinski, Tatum A. Jolink, Olivia C. Smith, Amit M. Ariely, Paschal Sheeran.

**Investigation:** Steven G. Buzinski, Tatum A. Jolink, Olivia C. Smith, Amit M. Ariely, Paschal Sheeran.

**Methodology:** Steven G. Buzinski, Tatum A. Jolink, Olivia C. Smith, Amit M. Ariely, Paschal Sheeran.

**Project administration:** Steven G. Buzinski, Tatum A. Jolink.

**Resources:** Steven G. Buzinski, Tatum A. Jolink.

**Software:** Steven G. Buzinski, Tatum A. Jolink, Olivia C. Smith, Amit M. Ariely.

**Supervision:** Steven G. Buzinski, Tatum A. Jolink.

**Writing – original draft:** Steven G. Buzinski, Tatum A. Jolink, Olivia C. Smith, Amit M. Ariely, Paschal Sheeran.

**Writing – review & editing:** Steven G. Buzinski, Tatum A. Jolink, Olivia C. Smith, Amit M. Ariely, Paschal Sheeran.

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
