## [Decision Letter · Decision Letter 0]

19 Dec 2025

Promoting Energy Conservation through Attentional Control and Construct Activation: A Field Test at a Campus Laundry

PLOS One

Dear Dr. Buzinski,

Thank you for submitting your manuscript to PLOS ONE. After careful consideration, we feel that it has merit but does not fully meet PLOS ONE’s publication criteria as it currently stands. Therefore, we invite you to submit a revised version of the manuscript that addresses the points raised during the review process.

We look forward to receiving your revised manuscript.

Kind regards,

Jamil Afzal, Ph.D, Post Doc

Academic Editor

PLOS One

Journal Requirements:

2. Please note that your Data Availability Statement is currently missing the repository name and/or the DOI/accession number of each dataset. If your manuscript is accepted for publication, you will be asked to provide these details on a very short timeline. We therefore suggest that you provide this information now, though we will not hold up the peer review process if you are unable.

3. We note that Figure 1a in your submission contain copyrighted images. All PLOS content is published under the Creative Commons Attribution License (CC BY 4.0), which means that the manuscript, images, and Supporting Information files will be freely available online, and any third party is permitted to access, download, copy, distribute, and use these materials in any way, even commercially, with proper attribution. For more information, see our copyright guidelines: http://journals.plos.org/plosone/s/licenses-and-copyright.

1. You may seek permission from the original copyright holder of Figure 1a to publish the content specifically under the CC BY 4.0 license.

Reviewers' comments:

Reviewer's Responses to Questions

**Comments to the Author**

1. Is the manuscript technically sound, and do the data support the conclusions?

Reviewer #1: Yes

Reviewer #2: Partly

2. Has the statistical analysis been performed appropriately and rigorously?

Reviewer #1: Yes

Reviewer #2: Yes

3. Have the authors made all data underlying the findings in their manuscript fully available?

Reviewer #1: Yes

Reviewer #2: Yes

4. Is the manuscript presented in an intelligible fashion and written in standard English?

Reviewer #1: Yes

Reviewer #2: Yes

Reviewer #1: This article provides a detailed discussion on how cold water washing can reduce energy consumption. After minor revisions, this paper can be published.

It is suggested to add corresponding charts for comparison.

Reviewer #2: Review

I commend the authors for addressing an important, practical issue. The increase in cold water use from 25% to 57% shows the impact of applied psychological science. The use of direct observation in both studies adds rigor to pro-environmental research. These findings are a valuable and scalable contribution.

Key Comments for Improvement

1. Theoretical Mechanism (Attention): The main weakness is that self-reported attention did not explain the intervention’s effect, even though the paper focuses on “Attentional Control.” You should discuss whether attention was just needed for noticing the messages, or if measurement issues hid a real effect. Please clarify why attention mattered but didn’t mediate the effect.

2. Study 2 Control Group Contamination: Not removing pre-existing signage in the control group threatens the study’s validity. This may have influenced the control group’s beliefs or behavior. Please discuss how this condition differs from a true baseline with no signage.

3. Low Normative Beliefs Scale Reliability: The reliability of the Normative Beliefs scale (α = .62) in Study 1 is low. Since the intervention used a strong normative message, this weak measurement may explain why you didn’t find an effect. Please note this as a limitation.

4. Limited Sample Generalizability: Over 90% of participants in both studies were first-year undergraduates, who have little laundry experience. State clearly that these results mainly apply to novice groups.

5. Need to Separate Intervention Elements: Future research should break down the intervention package (arrow + three messages) to see if just the arrow or just the control belief message is effective, making the intervention more efficient.

Minor Grammar and Clarity Issues:

page 11, line 233: Change to “The second was the self-reported question: ‘Which wash cycle did you select?’”

Page 10, line 217: Change “cutout window” to “large viewing window” or “large pane-glass window.”

**Do you want your identity to be public for this peer review?** For information about this choice, including consent withdrawal, please see our Privacy Policy

Reviewer #1: No

Reviewer #2: **Yes:** Aftab Haider

---

## [Author Response · Author response to Decision Letter 1]

1 Feb 2026

Response to Additional Requirements:

1. Please ensure that your manuscript meets PLOS ONE's style requirements, including those for file naming. The PLOS ONE style templates can be found at:

We have followed the PLOS ONE style requirements, and the manuscript is now in compliance.

2. Please note that your Data Availability Statement is currently missing the repository name and/or the DOI/accession number of each dataset. If your manuscript is accepted for publication, you will be asked to provide these details on a very short timeline. We therefore suggest that you provide this information now, though we will not hold up the peer review process if you are unable.

Repository name on OSF: Promoting Energy Conservation through Attention Control and Construct Activation: A Field Test at a Campus Laundry

Project DOI: https://doi.org/10.17605/OSF.IO/BFYWK

Study 1 DOI: https://osf.io/bfywk/files/osfstorage/604fc0e736e58b02a6bc24ea

Study 2 DOI: https://osf.io/bfywk/files/osfstorage/604fc6385ca6c700097880ea

3. We note that Figure 1a in your submission contain copyrighted images. All PLOS content is published under the Creative Commons Attribution License (CC BY 4.0), which means that the manuscript, images, and Supporting Information files will be freely available online, and any third party is permitted to access, download, copy, distribute, and use these materials in any way, even commercially, with proper attribution. For more information, see our copyright guidelines:http://journals.plos.org/plosone/s/licenses-and-copyright. We require you to either (1) present written permission from the copyright holder to publish these figures specifically under the CC BY 4.0 license, or (2) remove the figures from your submission.

Thank you for bringing this to our awareness. We have removed the copyrighted images.

Response to Reviewers:

I commend the authors for addressing an important, practical issue. The increase in cold water use from 25% to 57% shows the impact of applied psychological science. The use of direct observation in both studies adds rigor to pro-environmental research. These findings are a valuable and scalable contribution.

We thank the reviewer for the positive evaluation of the level of contribution of our research and their appreciation of the work’s translational value. We are grateful for the reviewer’s engaged and constructive feedback and have endeavored to address each of the points raised in the reviews.

Key Comments for Improvement

1. Theoretical Mechanism (Attention): The main weakness is that self-reported attention did not explain the intervention’s effect, even though the paper focuses on “Attentional Control.” You should discuss whether attention was just needed for noticing the messages, or if measurement issues hid a real effect. Please clarify why attention mattered but didn’t mediate the effect.

This is an important issue and we appreciate the reviewer’s request for greater clarity concerning the theoretical mechanism underlying the impact of our intervention. We have substantially revised the discussion to address the two points raised by the reviewer (pp. 20-21). First, we agree that a self-report measure of attention is suboptimal and that an ‘objective’ (i.e., device-based) measure would have afforded firmer conclusions about the role of attention. Second, the Attention-Readiness-Motivation framework indeed proposes that endogenous attention is a precursor of the impact of valuations but valuations are the proximal predictor of action. Our signage intervention simultaneously manipulated attention and valuation which makes it difficult to disentangle their effects. We point out that future studies should endeavor to manipulate attention and valuation in a factorial design to offer a stronger test of their individual and combined effects.

2. Study 2 Control Group Contamination: Not removing pre-existing signage in the control group threatens the study’s validity. This may have influenced the control group’s beliefs or behavior. Please discuss how this condition differs from a true baseline with no signage.

We should have been clearer about the nature of the control condition and thank the reviewer for bringing this issue to our attention. We have added a new paragraph to the method section in Study 2 (p. 15) and deleted reference to the control condition in the discussion. The new paragraph indicates that the control condition qualifies as a “treatment as usual” or “usual care” control and we reference Freedland et al.,’s (2019) seminal analysis of choice of intervention comparators. We point out that the laundry room signage required by the Housing Authority remained in place and no additional signage concerning cold-water laundering was provided. Thus, in our control condition, the laundry room was the same as it would have been if no intervention was tested, and was equivalent to the physical setting tested in Study 1.

3. Low Normative Beliefs Scale Reliability: The reliability of the Normative Beliefs scale (α = .62) in Study 1 is low. Since the intervention used a strong normative message, this weak measurement may explain why you didn’t find an effect. Please note this as a limitation.

We thank the Reviewer for this comment, as the lower scale reliability could indeed be one reason why normative beliefs were not correlated with experimental condition in Study 2 (r=.11). They were, we note, positively correlated with laundering with cold water, regardless of condition (r=.25). While speculative, another possibility for the lack of correlation between condition and normative beliefs is that the normative belief items focused on the importance or cleaning benefits of using cold water, whereas the normative message in the intervention was about fighting climate change. These different targeted normative beliefs may also explain the small correlation. We’ve noted the scale reliability as a limitation in the General Discussion on page 23.

4. Limited Sample Generalizability: Over 90% of participants in both studies were first-year undergraduates, who have little laundry experience. State clearly that these results mainly apply to novice groups.

Thank you for this comment. We have addressed it by adding the following language to page 23, “First, it may limit the generalizability of the results due to the demographics of the sample. A majority of the participants (94%) in the intervention study were first year university students. It is possible that this sample was relatively inexperienced in laundering, and thus our findings may be limited to novice groups.”

5. Need to Separate Intervention Elements: Future research should break down the intervention package (arrow + three messages) to see if just the arrow or just the control belief message is effective, making the intervention more efficient.

Thank you again. We agree with the reviewer and have addressed this point by adding the following language to page 23, “It was also the case that we were unable to decompose the influence of the two key components of our intervention – endogenous attention (i.e., the physical arrow and its placement) and psychological construct activation (i.e., the behavioral, normative, and control belief messages on the arrow). Factorial designs could be used in future research in order to formally test the relative, and interactive, influence of each.”

Minor Grammar and Clarity Issues:

page 11, line 233: Change to “The second was the self-reported question: ‘Which wash cycle did you select?’”

We have revised this line to reflect the reviewer’s language above.

Page 10, line 217: Change “cutout window” to “large viewing window” or “large pane-glass window.”

We have revised this line to, “large pane-glass window.”

---

## [Decision Letter · Decision Letter 1]

16 Feb 2026

Promoting Energy Conservation through Attention Control and Construct Activation: A Field Test at a Campus Laundry

PONE-D-25-45725R1

Dear Dr. Buzinski,

We’re pleased to inform you that your manuscript has been judged scientifically suitable for publication and will be formally accepted for publication once it meets all outstanding technical requirements.

Kind regards,

Jamil Afzal, Ph.D, Post Doc

Academic Editor

PLOS One

Reviewers' comments:

Reviewer's Responses to Questions

**Comments to the Author**

Reviewer #1: All comments have been addressed

Reviewer #2: All comments have been addressed

2. Is the manuscript technically sound, and do the data support the conclusions?

Reviewer #1: Yes

Reviewer #2: Yes

3. Has the statistical analysis been performed appropriately and rigorously?

Reviewer #1: Yes

Reviewer #2: Yes

4. Have the authors made all data underlying the findings in their manuscript fully available?

Reviewer #1: Yes

Reviewer #2: Yes

5. Is the manuscript presented in an intelligible fashion and written in standard English?

Reviewer #1: Yes

Reviewer #2: Yes

Reviewer #1: This has been revise and has become very good, I agree that the article was published in plos one. Good Lucky!

Reviewer #2: The authors have successfully addressed all previous comments. I am satisfied with the revisions and accept the manuscript.

**Do you want your identity to be public for this peer review?** For information about this choice, including consent withdrawal, please see our Privacy Policy

Reviewer #1: No

Reviewer #2: No

---

## [Editor Report · Acceptance letter]

PONE-D-25-45725R1

PLOS One

Dear Dr. Buzinski,

I'm pleased to inform you that your manuscript has been deemed suitable for publication in PLOS One. Congratulations! Your manuscript is now being handed over to our production team.

Kind regards,

on behalf of

Dr. Jamil Afzal

Academic Editor

PLOS One